# Autoregressive Models with Time-Dependent Coefficients—A Comparison between Several Approaches

**Rajae Azrak** [1] **and Guy Mélard** [2,*] 

1   Faculté des Sciences Juridiques Economiques et Sociales, Université Mohammed V—Rabat,
    Salé Route Outa Hssain, Sala Al Jadida, Salé B.P. 5295, Morocco
2   Solvay Brussels School of Economics and Management and ECARES, Université Libre de Bruxelles,
    CP 114/04, Avenue Franklin Roosevelt, 50, B-1050 Brussels, Belgium
*   Correspondence: guy.melard@ulb.be

**Abstract:** Autoregressive-moving average (ARMA) models with time-dependent (td) coefficients and marginally heteroscedastic innovations provide a natural alternative to stationary ARMA models. Several theories have been developed in the last 25 years for parametric estimations in that context. In this paper, we focus on time-dependent autoregressive (tdAR) models and consider one of the estimation theories in that case. We also provide an alternative theory for tdAR processes that relies on a $\rho$-mixing property. We compare these theories to the Dahlhaus theory for locally stationary processes and the Bibi and Francq theory, made essentially for cyclically time-dependent models, with our own theory. Regarding existing theories, there are differences in the basic assumptions (e.g., on derivability with respect to time or with respect to parameters) that are better seen in specific cases such as the tdAR(1) process. There are also differences in terms of asymptotics, as shown by an example. Our opinion is that the field of application can play a role in choosing one of the theories. This paper is completed by simulation results that show that the asymptotic theory can be used even for short series (less than 50 observations).

**Keywords:** nonstationary process; time series; time-dependent model; time-varying model; $\rho$-mixing property; locally stationary processes

## 1. Introduction

Autoregressive-moving average (ARMA) models with time-dependent (td) coefficients and marginally heteroscedastic innovations provide a natural alternative to stationary ARMA time series models. Several theories have been developed in the last 25 years for parametric estimations in that context (see [1]).

To simplify our presentation, before considering the autoregressive model of order $p$ or tdAR($p$), let us consider the case of the tdAR(1) model with a time-dependent coefficient $\phi_t^{(n)}$, which depends on time $t$ and also, possibly, on $n$, the length of the series. Marginal heteroscedasticity is introduced using another deterministic sequence $g_t^{(n)} > 0$. Let also $(e_t, t \in \mathbb{N})$ be a white noise process, consisting of independent random variables, not necessarily identically distributed, with a mean of zero and with a standard deviation $\sigma > 0$ and fourth-order cumulant $\kappa_{4t}$. The model is defined by

$$w_t^{(n)} = \phi_t^{(n)} w_{t-1}^{(n)} + g_t^{(n)} e_t. \tag{1}$$

The coefficient $\phi_t^{(n)}$ and $g_t^{(n)}$ depend on $t$ and sometimes, not always, on $n$, but also the parameters. We denote $\sigma_t^{(n)} = g_t^{(n)} \sigma$, the innovation standard deviation. For a given $n$, consider a sequence of observations $w^{(n)} = \left( w_1^{(n)}, w_2^{(n)}, \ldots, w_n^{(n)} \right)$ of the process. When $\phi_t^{(n)}$ or $g_t^{(n)}$ depend on $n$, we should speak of a triangular array process, not of a stochastic

process. Note that we use $g_t^{(n)}$ instead of $\left\{h_t^{(n)}\right\}^{1/2}$ in [1] to comply with the notations introduced in multivariate models (see [2]).

The AR(1) process with a time-dependent coefficient has been considered by Wegman, Tjøstheim, Kwoun and Yajima [3–5]. Hamdoune [6] and Dahlhaus [7] extended the results to autoregressive processes of order $p$. Azrak and Mélard [1], denoted AM (see Appendix A), and Bibi and Francq [8], denoted BF, have considered tdARMA processes. Contrarily to AM, in BF, the coefficients depend only on $t$, not on $n$. Additionally, although the basic assumptions of BF and AM are different, their asymptotics are somewhat similar but differ considerably from those of Dahlhaus [7] based on locally stationary processes (LSP), where the dependence on $t$ and $n$ is only through their ratio $t/n$ (see the nice overview by Dahlhaus in [9]). For simplicity, we will compare these approaches on autoregressive models.

Two approaches can be sketched for asymptotic theories within nonstationary processes (see [10]). Approach 1 consists of analyzing the behavior of the process when $n$ tends to infinity. That assumes some generating mechanism in the background that remains the same over time. Two examples can be mentioned: processes with periodically changing coefficients and cointegrated processes. It is in the former context that BF [8] have established asymptotic properties for parameter estimators in the case where $n$ goes to infinity. Approach 2 for asymptotics within nonstationary processes consists of determining how estimates that are obtained for a finite and fixed sample size behave. This is the setting for describing, in general, the properties of a test under local alternatives (where the parameter space is rescaled by $1/\sqrt{n}$) or in nonparametric regression. Approach 2 is the framework considered by Dahlhaus [7] for LSP that we will briefly summarize now. First, there is an assumption of local stationarity that imposes continuity with respect to time and even differentiability (although [9] tries to replace these assumptions with bounded variation alternatives). Additionally, $n$ is not simply increased to infinity. The coefficients, such as $\phi_t^{(n)}$, are considered as a function of rescaled time $t/n$. Therefore, everything happens as if the time is rescaled to the interval $[0;1]$. Suppose $\phi_t^{(n)} = \widetilde{\phi}_{t/n}$ and $g_t^{(n)} = \widetilde{g}_{t/n}$, where $\widetilde{\phi}_u$ and $\widetilde{g}_u$, $0 \leq u \leq 1$, depend on a finite number of parameters, are differentiable functions of $u$ and such that $\left|\widetilde{\phi}_u\right| < 1$ for all $u$. The model is written as

$$w_t^{(n)} = \widetilde{\phi}_{t/n} w_{t-1}^{(n)} + \widetilde{g}_{t/n} e_t. \tag{2}$$

As a consequence, the assumptions made in the LSP theory are quite different from those of AM and BF, for example, due to the different nature of the asymptotics. The AM approach is somewhere between these two approaches, 1 and 2, sharing parts of their characteristics but not all of them. In Section 2, we specialize the assumptions of AM for tdAR($p$) processes and consider the special case of a tdAR(1) process. In Appendix B, we provide an alternative theory for tdAR($p$) processes that relies on a $\rho$-mixing property. The AM theory is further illustrated in Appendix C by simulation results on a tdAR(2) process. The LSP and BF theories are summarized in Sections 3 and 4, respectively. In Section 5, we compare the Dahlhaus LSP theory with our own AM theory. This is partly explained thanks to examples. The differences in the basic assumptions are emphasized. Similarly, in Section 6, a comparison is presented between the AM and BF approaches, before the conclusions in Section 7.

## 2. The AM Theory for Time-Dependent Autoregressive Processes

Let us consider the AM theory in the special case of tdAR($p$) processes. We want also to see if simpler conditions can be derived for the treatment of pure autoregressive processes.

We consider a triangular array of random variables $w = \left(w_t^{(n)}, t = 1, \dots, n, n \in \mathbb{N}\right)$ defined on a probability space $\left(\Omega, F, P_\beta\right)$, with values in $\mathbb{R}$, whose distribution depends on a vector $\beta = (\beta_1, \dots, \beta_r)$ of unknown parameters to be estimated, with $\beta$ lying in an

open set $B$ of a Euclidean space $\mathbb{R}^r$. The true value of $\beta$ is denoted by $\beta^0$. By abuse of the language, we will nevertheless talk about the process $w$.

**Definition 1.** *The process $w$ is called an autoregressive process of order $p$, with time-dependent coefficients, if and only if it satisfies the equation*

$$w_t^{(n)} = \sum_{k=1}^{p} \phi_{tk}^{(n)} w_{t-k}^{(n)} + g_t^{(n)} e_t, \tag{3}$$

*where $(e_t, t \in \mathbb{N})$ and $g_t^{(n)}$ are as before.*

We denote again $\sigma_t^{(n)} = \sigma g_t^{(n)}$. The initial values $w_t$, $t < 1$ are supposed to be equal to zero. The $r$-dimensional vector $\beta$ contains all the parameters to be estimated, those in $\phi_{tk}^{(n)}$, $k = 1, \ldots, p$ and those in $g_t^{(n)}$, but not the scale factor $\sigma$. which is estimated separately. We suppose a specific deterministic parameterization in the function of $t$ and $n$. Let $\phi_{tk}^{(n)}(\beta)$ be the parametric coefficient with $\phi_{tk}^{(n)} = \phi_{tk}^{(n)}(\beta^0)$ and, similarly, $g_t^{(n)}(\beta)$ with $g_t^{(n)} = g_t^{(n)}(\beta^0)$. Let $e_t^{(n)}(\beta)$ be the residual for a given $\beta$:

$$e_t^{(n)}(\beta) = w_t^{(n)} - \sum_{k=1}^{p} \phi_{tk}^{(n)}(\beta) w_{t-k}^{(n)}. \tag{4}$$

Note that $e_t^{(n)}(\beta^0) = g_t^{(n)} e_t$.

Thanks to the assumption about the initial values and by using (3) recurrently, it is possible to write the pure moving average representation of the process:

$$w_t^{(n)} = \sum_{k=0}^{t-1} \psi_{tk}^{(n)} g_{t-k}^{(n)} e_{t-k} \tag{5}$$

(see [1] for a recurrence formula for the $\psi_{tk}^{(n)}$). Let $F_t$ be the $\sigma$-field generated by $\left(w_s^{(n)}, s \leq t\right)$ and, hence, by $(e_s, s \leq t)$, which explains why a superscript $^{(n)}$ is suppressed, and $F_0 = \{\varnothing, \Omega\}$. To simplify the presentation, we denote $E_{\beta^0}(\cdot(\beta)) = \{E_\beta((\beta))\}_{\beta=\beta^0}$ and, similarly, $\text{var}_{\beta^0}(\cdot)$ and $\text{cov}_{\beta^0}(\cdot)$. We are interested in the Gaussian quasi-maximum likelihood estimator:

$$\hat{\beta}^{(n)} = \text{argmin}_{\beta \in \mathbb{R}^r} \frac{1}{2} \sum_{t=1}^{n} \left\{ \log\left\{\sigma_t^{(n)}(\beta)\right\}^2 + \left(\frac{e_t^{(n)}(\beta)}{\sigma_t^{(n)}(\beta)}\right)^2 \right\}. \tag{6}$$

Denote $\alpha_t^{(n)}(\beta)$ the expression between the curved brackets in (6). Note that the first term of $\alpha_t^{(n)}(\beta)$ will sometimes be omitted, corresponding to a weighted least squares method, especially when $\sigma_t^{(n)}(\beta)$ does not depend on the parameters, or even ordinary least squares, when $\sigma_t^{(n)}(\beta)$ does not depend on $t$. BF considers that estimation method and, also, a variant where the denominator is replaced by a consistent estimator. Other estimators are also used in the LSP theory (see Section 3).

We need expressions for the derivatives of $e_t^{(n)}(\beta)$ with respect to $\beta$ using (4). The first derivative is

$$\frac{\partial e_t^{(n)}(\beta)}{\partial \beta_i} = -\sum_{k=1}^{p} \frac{\partial \phi_{tk}^{(n)}(\beta)}{\partial \beta_i} w_{t-k}^{(n)}, \quad i = 1, \ldots, r. \tag{7}$$

It will be convenient to write it as a pure moving average process using (5)

$$\frac{\partial e_t^{(n)}(\beta)}{\partial \beta_i} = \sum_{k=1}^{t-1} \psi_{tik}^{(n)}(\beta) g_{t-k}^{(n)} e_{t-k}, \tag{8}$$

for $i = 1, \ldots, r$, where the coefficients $\psi_{tik}^{(n)}(\beta)$ are obtained by the following relations:

$$\psi_{tik}^{(n)}(\beta) = -\sum_{u=1}^{\min(k,p)} \frac{\partial \phi_{tu}^{(n)}(\beta)}{\partial \beta_i} \psi_{t-u,k-u}^{(n)}.$$

Let $\psi_{tik}^{(n)} = \psi_{tik}^{(n)}(\beta^0)$ (with respect to [1], we improved the presentation in light of [8], especially by distinguishing $\beta$ and $\beta^0$. The notations for the innovations are also changed to emphasize that $F_t$ does not depend on $n$). Similarly, we introduce $\psi_{tijk}^{(n)}(\beta)$ and $\psi_{tijlk}^{(n)}(\beta)$ using the second and third derivatives of $e_t^{(n)}(\beta)$ for $i$, $j$, $l = 1, \ldots, r$ and define $\psi_{tijk}^{(n)} = \psi_{tijk}^{(n)}(\beta^0)$ and $\psi_{tijlk}^{(n)} = \psi_{tijlk}^{(n)}(\beta^0)$.

Under all the assumptions of Theorem 2′ of [1] (see Appendix A), the estimator $\hat{\beta}^{(n)}$ converges in probability to $\beta^0$ and $\sqrt{n}(\hat{\beta}^{(n)} - \beta^0) \xrightarrow{L} N(0, V^{-1}WV^{-1})$ when $n \to \infty$, where, with $^T$ denoting transposition,

$$W = \lim_{n \to \infty} \frac{1}{4n} \sum_{t=1}^{n} E_{\beta^0}\left(\frac{\partial \alpha_t^{(n)}(\beta)}{\partial \beta} \frac{\partial \alpha_t^{(n)}(\beta)}{\partial \beta^T}\right), \tag{9}$$

and

$$V = \lim_{n \to \infty} \frac{1}{2n} \sum_{t=1}^{n} E_{\beta^0}\left(\frac{\partial^2 \alpha_t^{(n)}(\beta)}{\partial \beta \partial \beta^T}\bigg| F_{t-1}\right), \tag{10}$$

where $\alpha_t^{(n)}(\beta)$ was defined after (6).

**Example 1.** *The tdAR(1) process:*
*Let us consider a tdAR(1) process defined by (1) with the parametric coefficient* $\phi_t^{(n)}(\beta) = \phi_{t1}^{(n)}(\beta)$, *with true value* $\phi_t^{(n)} = \phi_t^{(n)}(\beta^0)$. *For example, see (13) or (14) later, or* $\phi_t^{(n)}(\beta) = \beta_1 \sin(t + \beta_2)$ *as in [4]. We have, for* $\psi_{tk}^{(n)}$ *in (5):*

$$\psi_{tk}^{(n)} = \prod_{l=0}^{k-1} \phi_{t-l}^{(n)}, \ k = 1, \ldots, t-1,$$

*where a product for an empty set of indices is set to one. Note that, if* $\phi_t^{(n)}$ *is a constant* $\phi$, *then* $\psi_{tk}^{(n)} = \phi^k$, *such as for the MA representation of a stationary AR(1) process. Similarly,*

$$\psi_{tik}^{(n)}(\beta) = -\frac{\partial \phi_t^{(n)}(\beta)}{\partial \beta_i} \psi_{t-1,k-1}^{(n)} = -\frac{\partial \phi_t^{(n)}(\beta)}{\partial \beta_i} \prod_{l=1}^{k-1} \phi_{t-l}^{(n)},$$

*and analogous expressions for the second and third derivatives. The following is an application of Theorem 2′ of [1]. (Note that the assumption in [1] that the expectation of the fourth-order power of the variable* $w_t^{(n)}$ *is bounded is replaced by a bound for the sum over k of* $\left\{\psi_{tk}^{(n)}\right\}^2$ *).*

**Theorem 1.** *Consider a tdAR(1) process defined by (1) under the assumptions of Theorem A1 in Appendix A, except that* $H_{2′.1}$ *is replaced by* $H_{2′.1A}$:

$H_{2'.1A}$: *Let us suppose that there exist constants $C$, $\Psi$ $(0 < \Psi < 1)$, $M_1$, $M_2$ and $M_3$, such that the following inequalities hold for all $n$, $i$, $j$, $l$, and $k$ uniformly in $n$:*

$$\left| \psi_{tk}^{(n)} \right| < C\Psi^k, \quad \left| \left\{ \frac{\partial \phi_t^{(n)}(\beta)}{\partial \beta_i} \right\}_{\beta = \beta^0} \right| < M_1,$$

*and analogous bounds $M_2$ and $M_3$ for the second- and third-order derivatives.*

Then, the results of Theorem 1 are still valid.

**Proof.** Let us show the first of the inequalities in $H_{2'.1}$, since the others are similar. Consider for $\nu = 1, \ldots, t - 1$

$$\sum_{k=\nu}^{t-1} \left\{ \psi_{tik}^{(n)} \right\}^2 = \left( \frac{\partial \phi_t^{(n)}(\beta)}{\partial \beta_i} \right)_{\beta=\beta^0}^2 \sum_{k=\nu}^{t-1} \left\{ \psi_{t-1,k-1}^{(n)} \right\}^2 < \frac{M_1^2 C^2 (\Psi^2)^{\nu-1}}{1 - \Psi^2}.$$

Hence, $N_1 = M_1^2 C^2 (1 - \Psi^2)^{-1}$ and $\Phi = \Psi^2 < 1$. $\square$

**Remark 1.** *Note that the first inequality of $H_{2'.1A}$ is true when $\left| \phi_t^{(n)} \right| < 1$ for all $t$ and $n$, but this is not a necessity. A finite number of those $\phi_t^{(n)}$ can be greater than 1 without any problem. For example, $\phi_t^{(n)}(\beta) = 4(1 + \beta/n)(t/n)(1 - t/n)$, with $0 < \beta < 1$ would be acceptable, because the interval around $t/n = 0.5$, where the coefficient is greater than 1, shrinks when $n \to \infty$. With this in mind, Example 3 of [1] can be slightly modified to allow the upper bound of the $\left| \phi_t^{(n)} \right|$'s be greater than one. This will be illustrated in* Section 5. *Note also that the other inequalities of $H_{2'.1A}$ are easy to check.*

One of the assumptions of Theorem A1 in Appendix A, $H_{2'.6}$, is particularly strange at first sight, although it could be checked in the examples of Section 4 in [1]. It is interesting to note that, at least in the framework of autoregressive processes, it can be replaced by a more standard ρ-mixing condition. This is done in Appendix B. We were unfortunately not able to generalize it in the time-dependent moving average (MA) or ARMA models.

In [1], a few simulations were presented for simple tdAR(1) and tdMA(1) models, moreover when the generated series were stationary. It allowed us to assess empirically the theory but not to put it in jeopardy. In Appendix C, we show simulations for a tdAR(2) model, where the true coefficients $\phi_{t1}^{(n)}$ and $\phi_{t2}^{(n)}$ vary extremely. The parameterization used is

$$\phi_{tk}^{(n)}(\beta) = \phi_k' + \frac{1}{n-1}\left(t - \frac{n+1}{2}\right)\phi_k'', \quad k = 1, 2. \tag{11}$$

The results are good, although the model cannot be fitted on some of the generated series, especially for short series of a length $n = 50$. Except for the behavior at the end of the series, and with a small approximation of $t/(n-1)$ by $t/n$, it will be seen in Section 5 that the data generator process fulfills the assumptions of the LSP theory, so these simulations can also be seen as illustrative of that theory. It should be noted that there are few simulation experiments mentioned in the LSP literature. The word "simulation" is mentioned twice in [9] but each time to express a request.

Estimation of all tdAR models (and tdARMA models as well) is done by quasi-maximum likelihood estimation (QMLE) using numerical optimization of an exact Gaussian maximum likelihood function. An algorithm for the exact computation of the likelihood is based on a Cholesky factorization for band matrices [11], but an algorithm based on the Kalman filter [12] can be used as well. Since no software package can treat this, it has been

included in a specific program called ANSECH, written in Fortran and included in Time Series Expert (see [13]), which can be made available by the second author.

In Section 6 of [1], there are illustrative numerical examples of the AM theory. They are all based on Box–Jenkins series A, B and G (see [14]). These examples involve tdARMA models, not tdAR models. To illustrate a tdAR model, let us consider Box–Jenkins series D (see [14]). It is a series of 310 chemical process viscosity readings every hour. The fitted model was an AR(1) model with a constant. Here, we replaced the constant autoregressive coefficient with a linear function of time, as in (11) for $k = 1$ (but omitting the inverse of $n - 1$), and fitted a model on the first $n = 300$ observations. We used the exact likelihood function but (nonlinear) least squares would give nearly similar results because the series is long enough to remove the effect of the initial value at time $t = 0$. The results in Table 1 show the estimates for the three parameters and the corresponding standard errors obtained from the estimator of $V$ deduced from the optimization algorithm and justified by the asymptotic theory. Since the $t$-value for $\phi_1''$ is equal to $-2.7$, we can reject the null hypothesis of a constant coefficient $\phi_{t1}^{(n)}(\beta) = \phi_1$. Given these estimates, the coefficient $\phi_{t1}^{(n)}$ varies linearly between 0.9549 and 0.7307. This is not surprising, because if we compute the autocorrelation at lag 1 around time 90, we find 0.87, while around time 210, we find 0.75.

**Table 1.** Estimates of a (homoscedastic) tdAR(1) model defined by (1) for $p = 1$ with a constant (denoted MEAN 1). The parameters $\phi_1'$ and $\phi_1''$ are, respectively, denoted as AR 1 and TDAR 1.

| | **Final Values of the Parameters** | | | **With 95% Confidence Limits** | | |
|---|---|---|---|---|---|---|
| | Name | Value | Std Error | $t$-Value | Lower | Upper |
| 1 | MEAN 1 | 9.3223 | 0.10676 | 87.3 | 9.1 | 9.5 |
| 2 | AR 1 | 0.84640 | $2.97198 \times 10^{-2}$ | 28.5 | 0.79 | 0.90 |
| 3 | TDAR 1 | $-7.25476 \times 10^{-4}$ | $2.71992 \times 10^{-4}$ | $-2.7$ | $-1.26 \times 10^{-3}$ | $-1.92 \times 10^{-4}$ |

The AM theory was generalized to vector processes by [2], who treated the case of tdVARMA processes where the model coefficients did not depend on $n$, and by [15] for the general case, called tdVARMA$^{(n)}$ processes. Additionally, [16] provided a better foundation for the asymptotic theory for array processes, a theorem for a reduction of the order of moments from 8 to slightly more than 4 and tools for obtaining the asymptotic covariance matrix of the estimator. In [17], there was an example of vector tdAR and tdMA models on monthly log returns of IBM stock and the S&P500 index from January 1926 to December 1999, treated first in [18].

## 3. The Theory of Locally Stationary Processes

We gave in Section 1 some elements of the theory of Dahlhaus [7]. It is based on a class of locally stationary processes (LSP), which means a sequence of stationary processes, based on a stochastic integral representation:

$$w_t^{(n)} = \int_{-\pi}^{\pi} e^{i\lambda t} A_t^{(n)}(\lambda) d\xi(\lambda), \tag{12}$$

where $\xi(\lambda)$ is a process with independent increments and $A_t^{(n)}(\lambda)$ fulfills a condition to be called a slowly varying function of $t$. The theory will be well-adapted to time series that will be called locally stationary time series (LSTS).

In the case of autoregressive processes, which are emphasized in this paper, for example, an AR(1) process, that means that the observations around time $t$ are supposed to be generated by a stationary AR(1) process with some coefficient $\phi_t$. Stationarity implies that $-1 < \phi_t < 1$. Around time $t$, fitting is done using the process at time $t$. More generally, for AR($p$) processes, the autoregressive coefficients are such that the roots of the autoregressive polynomial are greater than 1 in the modulus.

The estimation method is based either on a spectral approach or a Whittle approximation of the Gaussian likelihood. The author of [19] also sketched out a maximum likelihood estimation method.

As mentioned above, the LSP approach of doing asymptotics relies on rescaling time $t$ in $u = t/n$. That does not mean that the process is considered in a continuous time, but at least that its coefficients are considered in a continuous time. Asymptotics is done by assuming an increasing number of observations between 0 and 1. That means that the coefficients are considered as a function of $t/n$ not separately as a function of $t$ and $n$. This is nearly the same as was assumed in (11) since $t/(n-1)$ is close to $t/n$ for large $n$. Note, however, that Example 1 of [1] is not in that class of processes. More generally, processes where the coefficients are periodic functions of $t$ are excluded from the class of processes under consideration. Of course, what was said about the coefficients is also valid for the innovation standard deviation. If the latter is a periodic function of time $t$, with a given period $s$, the process is not compatible with time rescaling. We will compare the LSP theory with the AM theory in Section 5.

That being said, the theory of LSPs has received considerable attention in the statistical literature. In his review [9], Dahlhaus listed a large number of extensions, generally by other authors, for univariate or multivariate models; for linear and nonlinear models and by parametric, semi-parametric and nonparametric methods. In particular, the following topics are overviewed, with several references for each of them: wavelet LSP, testing of LSP, in particular, testing for stationarity, bootstrap methods for LSP, model misspecification and model selection, likelihood theory and large deviations, recursive estimation, inference for a mean curve, piecewise constant models, long memory LSP, locally stationary random fields, discrimination analysis and applications in forecasting and finance. It is not useful to repeat a large number of references.

Furthermore, since [9], a large number of articles have appeared in the framework of the LSP theory, so many that it is only possible to mention a few of them. They are about a time-varying general dynamic factor model [20], time-varying additive models [21,22], nonparametric spectral analysis of multivariate series [23], bootstrapping [24], comparison of several techniques for identification of nonstationary multivariate autoregressive processes [25], inference for nonstationary time series autoregressions [26], the prediction of weakly LSP by autoregression [27], predictive inference for LSTS [28], frequency-domain tests for stationarity [29,30], cross-validations for LSP [31], adaptive covariance and spectrum estimation of multivariate LSP [32], large time-varying parameter VAR models by a nonparametric approach [33], a co-stationarity test of LSTS [34], towards a general theory of nonlinear LSPs [35], a quantile spectral analysis of LSTS [36], time-dependent dual-frequency coherence of a nonstationary time series [37] and nonparametric estimation of AR(1) LSP with periodicity [38].

Several examples illustrating these various methods for LSPs are included in these papers, such as in finance [21,26,36], environmental studies [22,28], biology with EEG signals [37], and also in economics with weekly egg prices [24].

## 4. The Theory of Cyclically Time-Dependent Models

Here, we will focus on BF (see [8]), but part of the discussion is also appropriate for older approaches such as [4–6]. In [8], Bibi and Francq developed a general theory of estimation for linear models with time-dependent coefficients, particularly aimed at the case of cyclically time-dependent coefficients (see also [39–42]).

The linear models include autoregressive but also moving average (MA) or ARMA models such as AM. The coefficients can depend on $t$ in a general way but not on $n$. Hence, $\phi_{tk}^{(n)}$ is written $\phi_{tk}$ in Definition 1. Heteroscedasticity is allowed similarly in the sense that the innovation variance can depend on $t$ (but not on $n$). The estimation method is a quasi-generalized least squares method.

The BF theory supports several classes of models. The periodic ARMA or PARMA models, where the coefficients are periodic functions of time, are an important class. Note

that the period does not need to be an integer. However, Section 3 of [8] also considered a switching model based on $\Delta$, a subset of integers in $\{1, 2, \ldots, n\}$ and its complement $\Delta^c$. For example, $\Delta$ can be associated with weekdays and $\Delta^c$ with the weekend. Then, the coefficient, e.g., $\phi_t$ in (1), depends on a parameter $\beta = (a, \widetilde{a})$ in the following way: $\phi_t = a1_\Delta(t) + \widetilde{a}1_{\Delta^c}(t)$, where $1_\Delta(t)$ denotes the indicator function, equal to 1 if $t$ belongs to $\Delta$ and 0 otherwise. Consequently, there are two different regimes. However, the composition of $\Delta$ and $\Delta^c$ can also be generated by an i.i.d. sequence of Bernoulli experiments, with some parameter $\pi$, provided they are independent of the white noise process $(e_t, t \in \mathbb{N})$.

Under appropriate assumptions, there is a theorem of almost sure consistency and convergence in the law of the estimator of $\beta$ to a normal distribution, somewhat as in Theorem A1 of Appendix A. Note that a strong consistency is proven here, not just convergence in probability. We will compare the BF theory with the AM theory in Section 6.

The BF approach has known several developments. In particular, it has been extended to time-dependent bilinear models [43], periodic bilinear models [44], periodic ARMA models (PARMA) with periodic bilinear innovations [45] and to GARCH models with time-varying coefficients [46,47]. More recent works on closely related models, such as weak ARMA models with regime changes [48], prefer to find a stationary and ergodic solution to the model equation. There are few examples in these papers, a remarkable exception being [47] with an application to daily gas spot prices.

## 5. A Comparison with the Theory of Locally Stationary Processes

In this section, we compare the AM approach described in Section 2 with the LSP approach described in Section 3. The basic model is (3), although the coefficients $\phi_{tk}^{(n)}$, $k = 1, \ldots, p$ and $g_t^{(n)}$ depend on $t$ and $n$ through $t/n$ only. Although LSP can be MA and ARMA processes (see [7]), the latter are rarely mentioned in the LSP literature. The bibliography of the overview paper [9] mentions "moving average", "MA" or "ARMA" only four times.

Dependency on $u = t/n$ of the model coefficients, as well as the innovation standard deviation, is assumed to be continuous and even differentiable in the initial LSP theory. We have mentioned that [9] suggested replacing that assumption with a bounded variation assumption (bounded variation functions are such that the derivative exists almost everywhere). In comparison, the other theories, including AM and BF (see [8]), accept discrete values of the coefficients according to time, without requiring a slow variation. They instead make assumptions of differentiability in the parameters.

Another point of discussion is as follows. To handle economic and social data with an annual seasonality, Box and Jenkins (see [14]) proposed the so-called seasonal ARMA processes, where the autoregressive and moving average polynomials are products of polynomials in the lag operator $B$ and polynomials in $B^s$ for some $s > 1$, for example, $s = 12$ for monthly data or $s = 4$ for quarterly data. Although the series generated by these stochastic processes are not periodic, with suitably initial values, they can show a seasonality with period $s$. Let us consider such ARMA processes with time-dependent coefficients, for example, a tdAR(12) defined by the equation $y_t = \phi_t^{(n)}(\beta)y_{t-12} + e_t$ with the same notations as in Section 1. There are exactly 11 observations between times $t$ and $t - 12$, and an increase in the total number of observations would not affect that. For such processes, Approach 1 of doing asymptotics, described in Section 1, seems to be the most appropriate, assuming that there is a larger number of years, not that there is a larger number of months within a year. Of course, Approach 2 of doing asymptotics is perfectly valid in all cases where the frequency of observation is more or less arbitrary.

To conclude, the AM approach is better suited for economic time series, where we can imagine that more years will become available (see the left-hand part of Figure 1). In other contexts, such as in biology and engineering, we can imagine that more data become available with an increased sampling rate (see the right-hand side of Figure 1). Then, the LSP theory seems more appropriate.

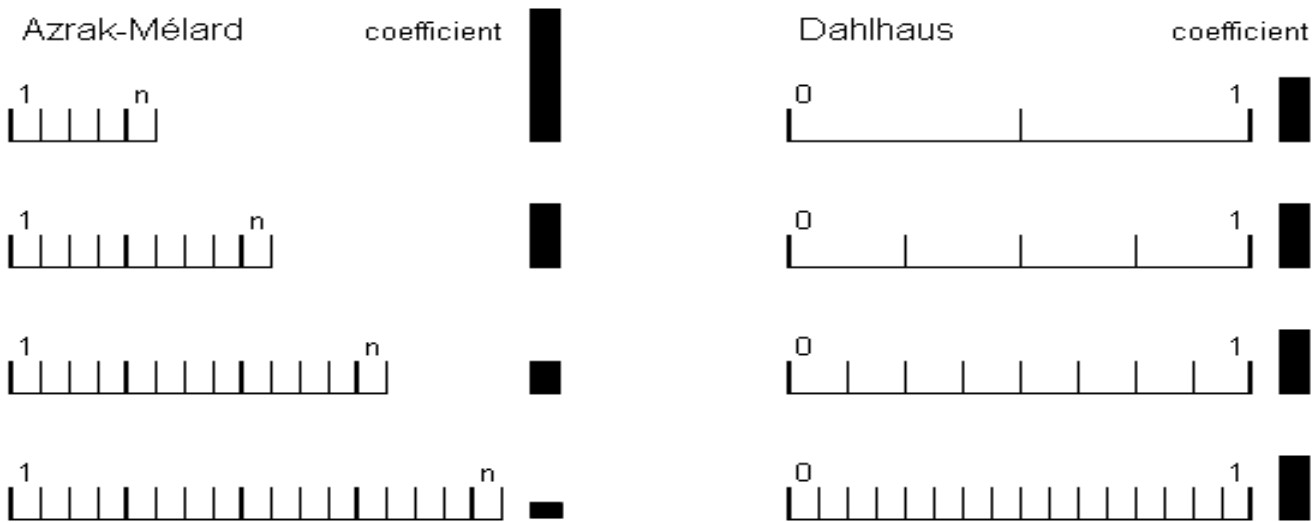

**Figure 1.** Schematic presentation on how to interpret asymptotics in AM and LSP theories (see the text for details).

In the following example, we will consider a tdAR(1) process but with the innovation standard deviation being a periodic function of time. Let us first show a unique artificial series of length 128 generated by (1) with

$$\phi_t^{(n)} = \phi' + \left(t - \frac{n+1}{2}\right)\phi'', \tag{13}$$

with $\phi' = 0.15$, $\phi'' = 0.015$ and the $e_t$ are normally and independently distributed with the mean 0 and variance $g_t$, where $g_t$ is a periodic function of $t$ with period 12, simulating seasonal heteroscedasticity for the monthly data. Furthermore, $g_t$, which does not depend on $n$, assumes values $\sqrt{g} = \sqrt{0.5}$ and $1/\sqrt{g} = \sqrt{2}$, each during six consecutive time points; hence, $g = 0.5$. We omitted the factor $1/(n-1)$ here since only one series length is considered. The series plotted in Figure 2 clearly shows a nonstationary pattern. The choices of $\phi' = 0.15$ and $\phi'' = 0.015$ are such that the autoregressive coefficient follows a straight line that goes slightly above $+1$ at the end of the series (see Figure 3). The parameters are estimated using the exact Gaussian maximum likelihood method. The representation of $g_t$ makes use of an old implementation for an intervention analysis for the innovation standard deviation [49]. The estimates (with the standard errors): $\hat{\phi}' = 0.0680$ ($\pm 0.0599$), $\hat{\phi}'' = 0.0155$ ($\pm 0.0013$) and $\hat{g} = 0.469$ ($\pm 0.0597$) are compatible with the true values. We provide the fit of $\phi_t^{(n)}$ and $g_t$, respectively, in Figures 3 and 4. Figures 5 and 6 give better insight into the relationships between the observations, broadly showing a negative autocorrelation during the first half of the series and a positive autocorrelation during the second half, as well as a small scatter during half of the year and a large scatter during the other half.

Note, finally, that this example is not compatible with the LSP theory, since $\phi_t^{(n)} > 1$ for some $t$, and $g_t$ being a piecewise constant is not a differentiable function of time. Additionally, the asymptotics related to that theory will be difficult to interpret, since $g_t$ is periodic with a fixed period.

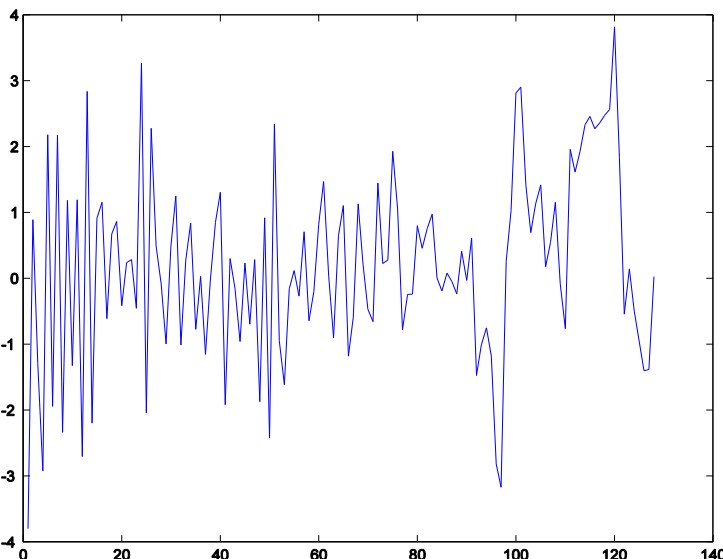

**Figure 2.** Artificial series produced using the process defined by (1) and (13) (see the text for details).

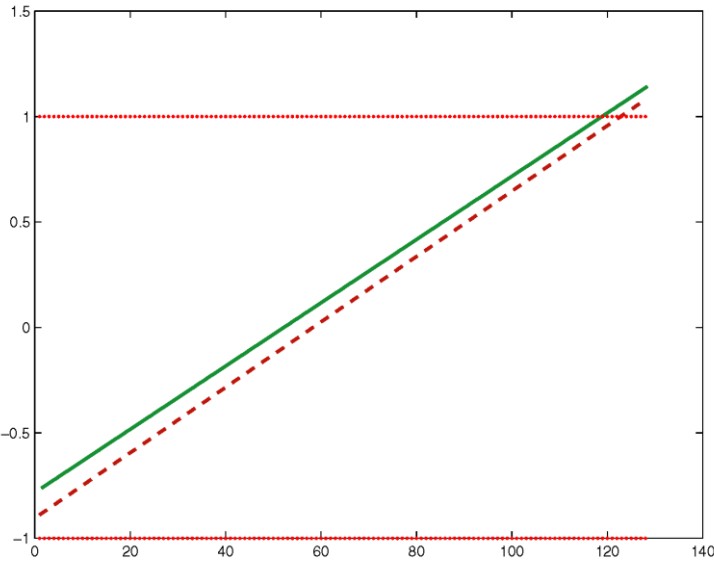

**Figure 3.** True values of $\phi_t^{(n)}$ (that go above 1!) (solid line) and their fit (discontinuous line).

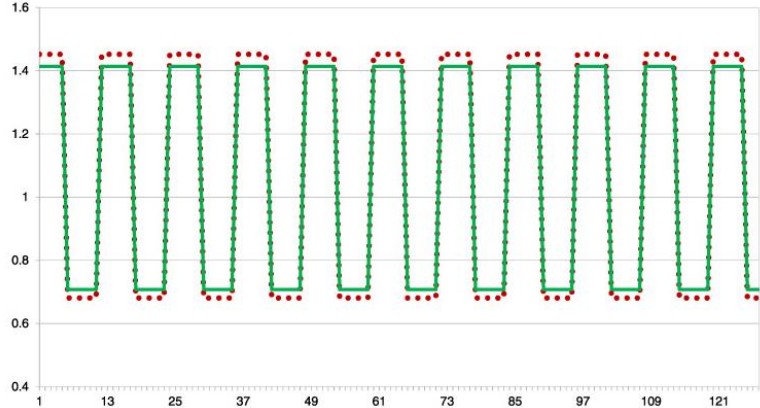

**Figure 4.** True value of $g_t$ (solid line) and their fit (discontinuous line).

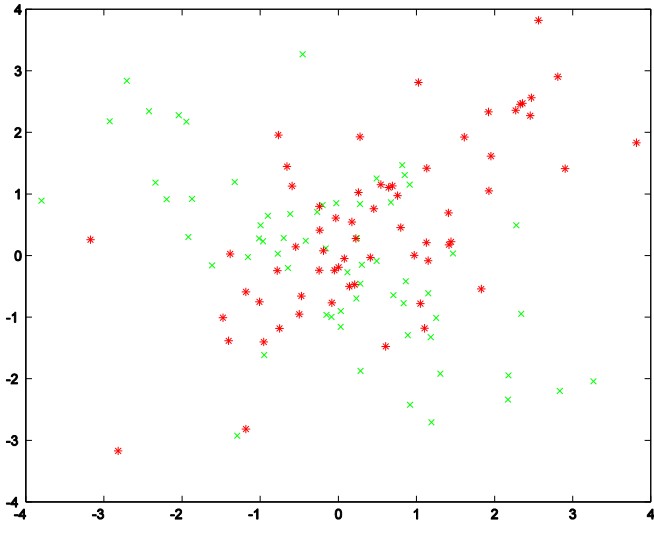

**Figure 5.** $w_t^{(128)}$ as a function of $w_{t-1}^{(128)}$ (crosses: $t \leq 64$, stars: $t > 64$).

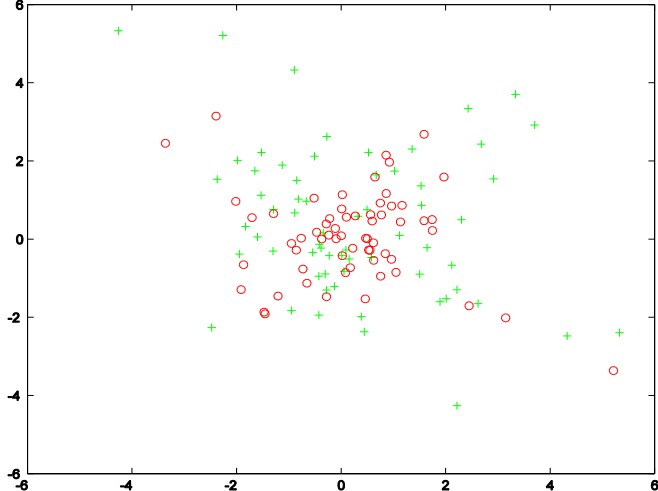

**Figure 6.** $w_t^{(128)}$ as a function of $w_{t-1}^{(128)}$ (plusses: high scatter, when $g_t = 2$; circles: small scatter, when $g_t = 0.5$).

We ran Monte Carlo simulations using the same setup, except that polynomials of two degrees were fitted for $\phi_t^{(n)}$ instead of a linear function of time. The parameterization is

$$\phi_t^{(n)}(\beta) = \phi' + \left( t - \frac{n+1}{2} \right) \phi'' + \left( t - \frac{n+1}{2} \right)^2 \phi''', \tag{14}$$

and $g_t$ is a periodic function that oscillates between the two values, $g$ and $1/g$, defined as above. The estimation program is the same as in AM but extended to cover polynomials of a time of degree up to three, as well as for AR (or similarly MA) coefficients, as for $g_t^{(n)}$. Estimates are obtained by numerically maximizing the exact Gaussian likelihood.

Some 1000 series of length 128 were generated using a program written in MATLAB with Gaussian innovations. Note that the estimation results were obtained for the 964 series only. They are provided in Table 2. Unfortunately, some estimates of the standard errors were unreliable, so their averages were useless and replaced by medians. The estimates of the standard errors are quite close to the empirical standard deviations. The fact is that the results are not as good as the simulation experiments described in Section 5 of [1], at least for a series of 100 observations or more, perhaps because the basic assumptions are only

barely satisfied with $\phi_t^{(n)}$ going nearly from about $-1$ to 1. In Table 3, we fitted the more adequate and simpler model with a linear function of time instead of a quadratic function of time. Then, the results were obtained for 999 series, and the estimated standard errors were always reliable so that their averages across the simulations are displayed.

**Table 2.** Theoretical values of the parameters, averages and standard deviations of the estimates across simulations and medians across simulations of the estimated standard errors $\phi'$ (true value: 0.15), $\phi''$ (true value: 0.015), $\phi'''$ (true value: 0) and $g$ (true value 0.5) for the tdAR(1) model described above for $n = 128$ and 964 replications (out of 1000).

| Parameter | | Standard | Median of |
|---|---|---|---|
| True Value | Average | Deviation | Standard Error |
| $\phi' = 0.15$ | 0.23554 | 0.14611 | 0.10380 |
| $\phi'' = 0.015$ | 0.01282 | 0.00222 | 0.00160 |
| $\phi''' = 0.0$ | $-0.00000$ | 0.00005 | 0.00005 |
| $g = 0.5$ | 0.54054 | 0.07857 | 0.08157 |

**Table 3.** Theoretical values of the parameters, averages and standard deviations of estimates across simulations, and averages across simulations of estimated standard errors $\phi'$ (true value: 0.15), $\phi''$ (true value: 0.015) and $g$ (true value 0.5) for the tdAR(1) model described above for $n = 128$ and 999 replications (out of 1000).

| Parameter | | Standard | Average of |
|---|---|---|---|
| True Value | Average | Deviation | Standard Error |
| $\phi' = 0.15$ | 0.22023 | 0.12577 | 0.06683 |
| $\phi'' = 0.015$ | 0.01305 | 0.00202 | 0.00146 |
| $g = 0.5$ | 0.54290 | 0.07847 | 0.06984 |

To end the section on a positive side, let us say that the three illustrative examples in Section 6 of [1] would give approximatively the same results under the LSP theory, provided that the same estimation method is used.

## 6. A Comparison with the Theory of Cyclically Time-Dependent Models

In this section, we compare the AM approach described in Section 2 with the BF approach described in Section 4. The basic model is (3) without the superscripts $^{(n)}$, since the coefficients $\phi_{tk}$, $k = 1, \ldots, p$ and $g_t$ do not depend on $n$. Therefore, it is clear that there is no sense to compare the LSP and the BF theories, which act on disjoint classes of processes.

We mentioned in Section 4 a few classes of processes to which the BF theory is applicable. The periodic ARMA or PARMA processes are surely compatible with the AM theory, even with an irrational period for the periodic functions of time (see also [2]). On the contrary, the switching model based on an i.i.d. sequence of Bernoulli experiments is not a particular case of the models treated in [1]. This is characteristic of the BF theory that the two other theories cannot handle at first sight. When the switching model is based on a fixed subset of integers $\Delta$, the AM theory can be adapted, especially in the weekdays versus weekend example. On the other hand, Examples 2–5 of [1] are incompatible with the BF theory, since the coefficients depend on $n$.

The basic assumptions of BF are different from those of AM. A comparison is difficult here, but it is interesting to note a less restrictive assumption of the existence of fourth-order moments, not eighth-order as in AM. Note that [16] has removed that requirement for the AM theory. Note that the expression for $I$ in [8], which corresponds to our $W$ in (9), did not involve fourth-order moments since no parameter was involved in the heteroscedasticity.

The process considered in Section 5 was an example for which the theory of locally stationary processes would not apply because of the periodicity in the innovation standard

deviation. That process is also an example for which the BF theory would not apply, because the autoregressive coefficient is a function of $t$ and $n$, not only of $t$.

## 7. Conclusions

This paper was motivated by suggestions to see if the results in [1] simplify much in the case of autoregressive or even tdAR(1) processes and by requests to compare more deeply the AM approach with others and push it in harder situations. We recalled the main result in Appendix A.

We showed that there are not many simplifications for tdAR processes, perhaps due to the intrinsically complex nature of ARMA processes with time-dependent coefficients. Nevertheless, we were able to simplify one of the assumptions for tdAR(1) processes. We took the opportunity of this study on autoregressive processes with time-dependent coefficients to develop in Appendix B an alternative approach based on a $\rho$-mixing condition instead of the strange assumption $H_{2'.6}$ made in AM. At least we could check that assumption in some examples, which was not the case for the mixing condition at present. Note that a mixing approach was the first we tried, before preferring $H_{2'.6}$. The latter could be extended to the tdMA and tdARMA processes, which was not the case for the mixing condition. Although the theoretical results for tdAR(2) processes could not be shown in closed-form expressions, the simulations in Appendix C indicate that the method is robust when causality becomes questionable. We showed more stressing simulations than in AM. ARIMA models could have been possible for these simulations and examples, but this paper focused on autoregressive processes. Practical examples of tdARMA models were already given in [1,8].

We also compared the AM approach to others, more especially Dahlhaus LSP theory [7] and the BF approach [8], aimed at cyclically time-dependent linear processes. Let us comment on this more deeply.

As in the LSP theory, a different process is considered by AM for each $n$. There are, however, several differences between the two approaches: (a) AM can cope with periodic evolutions with a fixed period, either in the coefficients or in the variance. (b) AM does not assume differentiability with respect to time but will with respect to the parameters; (c) to compensate, AM makes other assumptions that are more difficult to check, (d) which may explain why the LSP theory is more widely applicable: other models than just ARMA models and other estimation methods than the maximum likelihood, even semi-parametric methods, the existence of a LAN approach, etc. (e) AM is purely time domain-oriented, whereas the LSP theory is based on a spectral representation. An example with an economic inspiration and its associated simulation experiments showed that some of these assumptions of AM are less restrictive, but there is no doubt that others are more stringent. In our opinion, the field of applications can influence the kind of asymptotics. The Dahlhaus LSP approach is surely well-adapted to signal measurements in biology and engineering, where the sample span of time is fixed and the sampling interval is more or less arbitrary. This is not true in economics and management, where (a) time series models are primarily used to forecast a flow variable like sales or production, obtained by accumulating data over a given span of time, a month or a quarter, so (b) that the sampling period is fixed, and (c) moreover, some degree of periodicity is induced by seasonality. Here, it is difficult to assume that more observations become available during one year without strongly affecting the model. For that reason, even if the so-called seasonal ARMA processes, which are nearly the rule for economic data, are formally special cases of locally stationary processes, the way of doing asymptotics is not adequate. For the same reason, rescaling time is not natural when the coefficients are periodical functions of time.

Going now to a comparison of AM with the BF approach mainly aimed at cyclically time-dependent linear processes, we see the first fundamental difference is the fact that a different process is considered for each $n$ in AM, not in BF. That assumption of dependency on $n$, as well as on $t$, was introduced to be able to do asymptotics in cases that would not have been possible otherwise (except in adopting the Dahlhaus approach, of course) but, at

the same time, making it possible to represent a periodic behavior. When the coefficients are only dependent on $t$, not on $n$, the AM and BF approaches come close in the sense that (a) the estimation methods are close; (b) the assumptions are quite similar. The example shown to distinguish AM from LSP is also illuminating the differences between AM and BF. There remains that the switching model based on an i.i.d. Bernoulli process is not feasible in the AM approach.

In some sense, AM can be seen as partly taking some features of both the LSP and BF approaches. Some features, like the periodicity of the innovation variance, can be handled well in BF, while others, like slowly time-varying coefficients, are in the scope of LSP. However, a cyclical behavior of some innovation variance and slowly varying coefficients together (or the contrary: a cyclical behavior of some coefficients and slowly varying innovation variance) are not covered by Dahlhaus and BF theories but are by AM. The example in Section 5 may look artificial but includes all the characteristics that are not covered well by locally stationary processes and the corresponding asymptotic theory. It includes a time-dependent first-order autoregressive coefficient $\phi_t^{(n)}$ which is very realistic for an I(0) (i.e., not integrated) economic time series and an innovation variance $\sigma_t^2$, which is a periodic function of time (this can be explained by seasonality, as in a winter/summer effect). To emphasize the differences with the LSP approach, we assumed that $\phi_t^{(n)}$ goes slightly outside of the causality (or stationarity, in Dahlhaus terminology) region for some time and that $\sigma_t^2$ is piecewise-constant and, hence, not compatible with the differentiability at each time.

One aspect was not discussed in this paper: how to specify the time dependence of the model coefficients. For the example of Box and Jenkins series D, we mentioned the computation of autocorrelations on parts of the series. Otherwise, our examples were based on polynomial representations with respect to time, and we used tests to possibly reduce the degree of the polynomial. Other parameterizations than polynomials can be considered.

**Author Contributions:** Conceptualization, R.A. and G.M.; methodology, R.A. and G.M.; software, G.M.; validation, R.A. and G.M.; formal analysis, R.A. and G.M.; investigation, R.A. and G.M.; data curation, R.A. and G.M.; writing—original draft preparation, R.A. and G.M.; writing—review and editing, R.A. and G.M.; visualization, R.A. and G.M. and supervision, R.A. and G.M. All authors have read and agreed to the published version of the manuscript.

**Funding:** This research received no external funding.

**Institutional Review Board Statement:** Not applicable.

**Informed Consent Statement:** Not applicable.

**Data Availability Statement:** Only published and simulated data were used.

**Acknowledgments:** We thank the two anonymous reviewers of the first version who made very useful suggestions. We thank those who have made comments on a previous version of this paper, including Christian Francq, Marc Hallin, and mainly Rainer Dahlhaus and Denis Bosq. We thank the four reviewers of this version, who contributed to improving this paper.

**Conflicts of Interest:** The authors declare no conflict of interest.

## Appendix A. Theorem A1 (Theorem 2′ of [1])

Consider an autoregressive-moving average process with time-dependent coefficients (tdARMA), and suppose that the functions $\phi_{tk}^{(n)}(\beta)$, $\theta_{tk}^{(n)}(\beta)$ and $g_t^{(n)}(\beta)$ are three times continuously differentiable with respect to $\beta$ in the open set $B$ containing the true value $\beta^0$ of $\beta$ and that positive constants exist: $\Phi < 1$, $N_1, N_2, N_3, N_4, N_5, N_6, K_1, K_2, K_3, m, M, m_1$ and $K$, such that $\forall t = 1, \ldots, n$ uniformly with respect to $n$:

$$H_{2'.1} \quad \sum_{k=\nu}^{t-1} \left\{ \psi_{tik}^{(n)} \right\}^2 < N_1 \Phi^{\nu-1}, \quad \sum_{k=\nu}^{t-1} \left\{ \psi_{tik}^{(n)} \right\}^4 < N_2 \Phi^{\nu-1},$$

$$\sum_{k=\nu}^{t-1} \left\{ \psi_{tijk}^{(n)} \right\}^2 < N_3 \Phi^{\nu-1}, \ \sum_{k=\nu}^{t-1} \left\{ \psi_{tijk}^{(n)} \right\}^4 < N_4 \Phi^{\nu-1},$$

$$\sum_{k=1}^{t-1} \left\{ \psi_{tijlk}^{(n)} \right\}^2 < N_5, \ \sum_{k=1}^{t-1} \left\{ \psi_{tk}^{(n)} \right\}^2 < N_6, \ \nu = 1, \ldots, t-1, \ i, j, l = 1, \ldots, r;$$

$$H_{2'.2} \ \left| \left\{ \frac{\partial g_t^{(n)2}(\beta)}{\partial \beta_i} \right\}_{\beta=\beta^0} \right| \le K_1, \ \left| \left\{ \frac{\partial^2 g_t^{(n)2}(\beta)}{\partial \beta_i \partial \beta_j} \right\}_{\beta=\beta^0} \right| \le K_2,$$

$$\left| \left\{ \frac{\partial^3 g_t^{(n)2}(\beta)}{\partial \beta_i \partial \beta_j \partial \beta_l} \right\}_{\beta=\beta^0} \right| \le K_3 \ i, j, l = 1, \ldots, r;$$

$$H_{2'.3} \ 0 < m \le g_t^{(n)2} \le m_1;$$

$$H_{2'.4} \ E\left( e_t^{4+\delta} \right) \le K, \ \delta > 0.$$

Suppose furthermore that

$$H_{2'.5} \ \lim_{n\to\infty} \frac{1}{n} \sum_{t=1}^{n} \left[ \sigma^{-2} E_{\beta^0} \left( \frac{\partial e_t^{(n)}(\beta)}{\partial \beta_i} \left\{ g_t^{(n)}(\beta) \right\}^{-2} \frac{\partial e_t^{(n)}(\beta)}{\partial \beta_j} \right) \right.$$

$$\left. + \frac{1}{2} \left\{ \frac{\partial g_t^{(n)2}(\beta)}{\partial \beta_i} \left\{ g_t^{(n)}(\beta) \right\}^{-4} \frac{\partial g_t^{(n)2}(\beta)}{\partial \beta_j} \right\}_{\beta=\beta^0} \right] = V_{ij},$$

$i, j = 1, \ldots, r$, where the matrix $V = (V_{ij})_{1 \le i,j \le r}$ is a strictly definite positive matrix:

$$H_{2'.6} \ \frac{1}{n^2} \sum_{d=1}^{n-1} \sum_{t=1}^{n-d} \sum_{k=1}^{t-d} \psi_{tik}^{(n)} \left( \beta^0 \right) \psi_{t+d,j,k+d}^{(n)} \left( \beta^0 \right) g_{t-k}^{(n)2} \left( \beta^0 \right) = O\left( \frac{1}{n} \right), \text{ and}$$

$$\frac{1}{n^2} \sum_{d=1}^{n-1} \sum_{t=1}^{n-d} \sum_{k=1}^{t-1-d} \psi_{tik}^{(n)} \left( \beta^0 \right) \psi_{tjk}^{(n)} \left( \beta^0 \right) \psi_{t+d,i,k+d}^{(n)} \left( \beta^0 \right) \psi_{t+d,j,k+d}^{(n)} \left( \beta^0 \right) g_{t-k}^{(n)4} \left( \beta^0 \right) \kappa_{4,t-k} = O\left( \frac{1}{n} \right),$$

and where $\kappa_{4,t}$ is the fourth-order cumulant of $e_t$. Then, when $n \to \infty$,

- there exists an estimator $\hat{\beta}_n$, such that $\hat{\beta}^{(n)} \to \beta^0$ in probability;
- $n^{\frac{1}{2}} \left( \hat{\beta}^{(n)} - \beta^0 \right) \xrightarrow{L} N\left( 0, V^{-1}WV^{-1} \right)$, where there exists a matrix $W$ whose elements are defined by (9).

## Appendix B. Alternative Assumptions under a Mixing Condition

In this appendix, we shall need the processes to satisfy a mixing condition. The definition we use, e.g., [50], proposed by Kolmogorov and Rozanov [51] in the context of stationary processes, is the $\rho$-mixing condition.

**Definition A1.** *Let $(w_t, t \in Z)$ be a process (not necessarily stationary) of random variables defined on a probability space $(\Omega, F, P)$. We say that the process is $\rho$-mixing if there exists a sequence of positive real numbers $(\rho(d), d > 1)$, such that $\rho(d) \to 0$ as $n \to \infty$, where*

$$\rho(d) = \sup_{t \in \mathbb{Z}} \sup_{\substack{U \in \mathcal{L}^2(F_{-\infty}^t) \\ V \in \mathcal{L}^2(F_{t+d}^\infty)}} |\mathrm{corr}(U, V)|, \tag{A1}$$

*$F_{-\infty}^t$ is the $\sigma$-field spanned by $(w_s, s \le t)$, and $F_{t+d}^\infty$ is the $\sigma$-field spanned by $(w_s, s \ge t+d)$. Then, $\rho(d)$ is called the $\rho$-mixing coefficient of the process.*

Of course, if the process is strictly stationary, the supremum over $t$ disappears, and the definition coincides with the standard definition. The definition easily extends in our case of a triangular array process, since the $\sigma$-fields are generated by the innovations (and innovations in reverse time).

**Lemma A1.** *Let $(w_t, t \in \mathbb{Z})$ be a process (not necessarily stationary) that satisfies the $\rho$-mixing condition. Let a random variable $U \in \mathcal{L}^2(F_{-\infty}^t)$ and a random variable $V \in \mathcal{L}^2(F_{t+d}^\infty)$; then,*

$$\mathrm{cov}(U, V) \leq \rho(d)\{\mathrm{var}(U)\mathrm{var}(V)\}^{1/2}. \tag{A2}$$

This is obvious when taking into account (A1) (see [52]).

**Theorem A1.** *Consider a pure autoregressive process under the assumptions of Theorem 1, except that $H_{2'.6}$ in Theorem A1 is replaced by $H_{2'.6A}$:*
*$H_{2'.6A}$ : For $\beta = \beta^0$, let the process be $\rho$-mixing with mixing coefficient $\rho(d)$ bounded by an exponentially decreasing function, such that $|\rho(d)| < \rho^d$, with $0 < \rho < 1$.*

Then, the results of Theorem A1 are still valid.

**Proof.** $H_{2'.6}$ is used to prove two assumptions, $H_{1'.3}$ and $H_{1'.5}$ of Theorem 1' in [1], but the former is more demanding. We have to show (see Equation (A.13) of that paper) that

$$\frac{1}{n^2} \sum_{d=1}^{n-1} \sum_{t=1}^{n-d} \mathrm{cov}_{\beta^0}\left( \frac{\partial e_t^{(n)}(\beta)}{\partial \beta_i} \left\{ g_t^{(n)}(\beta) \right\}^{-2} \frac{\partial e_t^{(n)}(\beta)}{\partial \beta_j}, \frac{\partial e_{t+d}^{(n)}(\beta)}{\partial \beta_i} \left\{ g_{t+d}^{(n)}(\beta) \right\}^{-2} \frac{\partial e_{t+d}^{(n)}(\beta)}{\partial \beta_j} \right) \tag{A3}$$

is $O(1/n)$. We decompose the external sum in two sums, one for $d = 1, \ldots, p$ and one for $d = p + 1, \ldots, n - 1$, and we will show that both sums are $O(1/n)$. Using Cauchy–Schwarz inequality and the fact that the proof of Theorem 2 in [1] has shown that

$$\mathrm{E}_{\beta^0}\left( \frac{\partial e_t^{(n)}(\beta)}{\partial \beta_i} \left\{ g_t^{(n)}(\beta) \right\}^{-2} \frac{\partial e_t^{(n)}(\beta)}{\partial \beta_j} \right)^2$$

is bounded, uniformly in $t$, using only $H_{2'.1} - H_{2'.5}$, the first sum is indeed $O(1/n)$. $\square$

The general term of the second sum can be written as $\left\{ g_t^{(n)} \right\}^{-2} \left\{ g_{t+d}^{(n)} \right\}^{-2} H_{t,i,j,d}^{(n)}$, where

$$H_{t,i,j,d}^{(n)} = \mathrm{cov}_{\beta^0}\left( G_{t,i}^{(n)}(\beta) G_{t,j}^{(n)}(\beta), G_{t+d,i}^{(n)}(\beta) G_{t+d,j}^{(n)}(\beta) \right),$$

and $G_{t,i}^{(n)}(\beta) = \partial e_t^{(n)}(\beta)/\partial \beta_i$. Given (7), $U = G_{t,i}^{(n)} G_{t,j}^{(n)} \in \mathcal{L}^2(F_{-\infty}^t)$ and, provided $d > p$, $V = G_{t+d,i}^{(n)} G_{t+d,j}^{(n)} \in \mathcal{L}^2\left( F_{t+d-p}^\infty \right)$, for all $t$ and all $i$ and $j$. Indeed, the right-hand sides have finite variances by application of Cauchy–Schwarz inequality, and using the fact that $\mathrm{E}_{\beta^0}\left( G_{t,i}^{(n)}(\beta) \right)^4 \leq m_1^2\left( N_2 K^{1/2} + 3N_1 \right)\sigma^4$, using $H_{2'.1}$ and $H_{2'.3}$, uniformly in $n$ (see Equation A.9 in [1]). Additionally, $\left\{ g_t^{(n)} \right\}^{-2} \left\{ g_{t+d}^{(n)} \right\}^{-2} \leq m^{-2}$, using $H_{2'.3}$. By $H_{2'.6A}$ and using Lemma A1, $H_{t,i,j,d}^{(n)}$ is bounded by

$$\rho(d-p)\left\{ \mathrm{E}_{\beta^0}\left( G_{t,i}^{(n)}(\beta) G_{t,j}^{(n)}(\beta) \right)^2 \right\}^{1/2} \left\{ \mathrm{E}_{\beta^0}\left( G_{t+d,i}^{(n)}(\beta) G_{t+d,j}^{(n)}(\beta) \right)^2 \right\}^{1/2}.$$

That expression is uniformly bounded with respect to $t$. Since $H_{2'.6A}$ implies $\sum_{d=p+1}^{n-1} \left| \rho(d-p) \right| \leq \sum_{d=p+1}^{n-1} \rho^{d-p} < \infty$, (A3) is $O(1/n)$. The argument is similar to checking $H_{1'.5}$, but the expression to consider is

$$\frac{1}{n^2} \sum_{d=1}^{n-1} \sum_{t=1}^{n-d} \text{cov}_{\beta^0} \left( K_t^{(n)i}(\beta) \frac{\partial e_t^{(n)}(\beta)}{\partial \beta_i}, K_{t+d}^{(n)j}(\beta) \frac{\partial e_{t+d}^{(n)}(\beta)}{\partial \beta_j} \right),$$

where

$$K_t^{(n)i}(\beta) = 4 \frac{E\left\{ e_t^{(n)}(\beta) \right\}^3}{\sigma^4 g_t^{(n)6}(\beta)} \left\{ \frac{\partial g_t^{(n)2}(\beta)}{\partial \beta_i} \right\}.$$

The proof continues as in Theorem 2 and 2' of [1], using a weak law of large numbers for a mixingale array in [53] and referring to Theorems 1 and 1' of [1], which make use of a central limit theorem for a martingale difference array (see [54]), modified with a Lyapunov condition (see [55]).

**Remark A1.** *Strong mixing should be a nice requirement. However, on the one hand, even stationary AR(1) processes can be non-strong mixing, and, on the other hand, the covariance inequalities that are implied are not applicable in our context without stronger assumptions.*

**Remark A2.** *In an earlier version, uniformly strong mixing or $\varphi$-mixing was used. However, as Bosq in [50] pointed out, for Gaussian stationary processes, $\varphi$-mixing implies m-dependence for some m. Therefore, the AR processes should behave similar to MA processes, leaving just white noise. Finally, we opted for $\rho$-mixing. There were results for the stationary linear processes [56] and ARMA processes [57] but none for the non-stationary processes considered here. In practice, even if the $\rho$-mixing condition is more appealing, checking $H_{2'.6A}$ is more challenging than checking $H_{2'.6}$. For instance, in Example 3 of [1], it is possible to check $H_{2'.6}$.*

**Appendix C. tdAR(2) Monte Carlo Simulations**

The purpose of this appendix is to illustrate the procedure described in Section 2 on further, more stressing, simulation experiments than in [1]. In that paper, Monte-Carlo simulations were shown for nonstationary AR(1) and MA(1) models, with a time-dependent coefficient and a time-dependent innovation variance for several series lengths between 25 and 400 to show convergence empirically. The purpose was mainly to illustrate the theoretical results for these models, particularly the derivation of the asymptotic standard errors, and investigate the sensibility of the innovation distribution on the conclusions.

Here, we consider the tdAR(2) models described by (3) and (11) in nearly the same setup as in AM, except that the innovation variance is assumed constant, but the series are generated using a process with linearly time-dependent coefficients, not stationary processes. Only Gaussian innovations are simulated, so the inverse of V in (10) is used to produce standard errors. Since we are only interested in autoregressive models, it does not seem necessary to compare the exact maximum likelihood and the approximate or conditional maximum likelihood methods. Numerical optimization was used.

In the parametrization in (11), the two coefficients $\phi_{t1}^{(n)}$ and $\phi_{t2}^{(n)}$ vary with $t$ between $-0.5$ and $0.5$ for the former and between $-0.9$ and $0.5$ for the latter. If we consider the roots of the polynomials, $1 - \phi_{t1}^{(n)}z - \phi_{t2}^{(n)}z^2$, for the different $t$, that means they are complex until well after the middle of the series, where their modulus is large (about 8), whereas it is close to 1 at the beginning, and the smallest root is equal to 1 at the end of the series, so at the causality frontier. This is illustrated in Figure A1. It will therefore not be surprising if the empirical results are not as bright as in [1]. A plot of a sample series is shown in Figure A2, which illustrates the behavior: complex roots at the beginning correspond to oscillations, and a root close to 1 at the end corresponds to a strong positive autocorrelation.

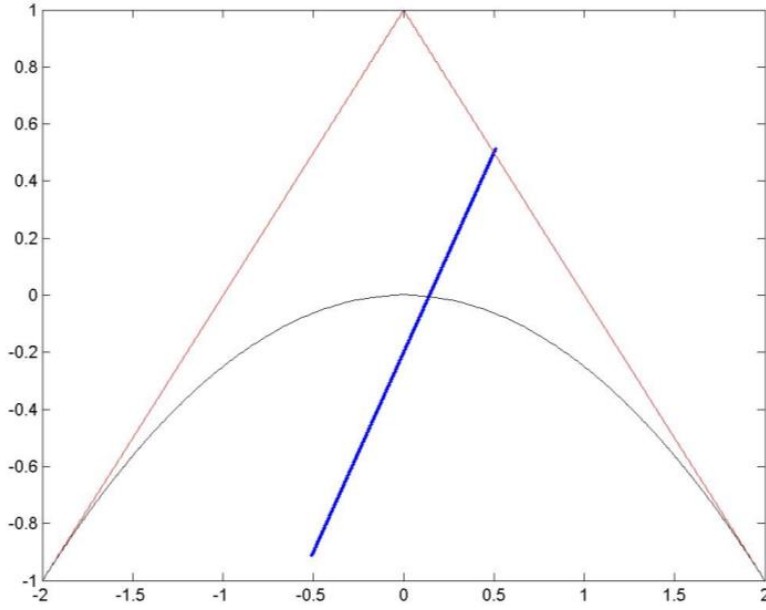

**Figure A1.** Variations of $\phi_{t1}^{(n)}$ (horizontal) and $\phi_{t2}^{(n)}$ (vertical) with respect to time t for $n = 50$; inside the triangle corresponds to the causality condition, and the curve separates complex roots (below) from real roots (above).

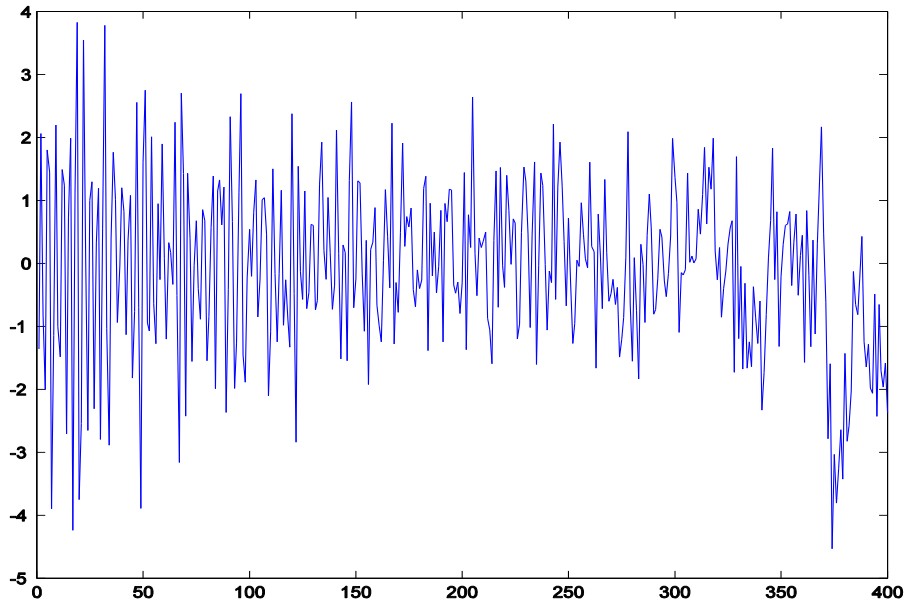

**Figure A2.** Plot of the data for one of the simulated tdAR(2) series for $n = 400$.

Table A1 for $n = 400$ shows that the estimates are close to the true values of the parameters and that the asymptotic standard errors are well-estimated since the average of these estimates agrees more or less with the empirical standard deviation. When $n = 50$, note, however, by comparison of the last two columns of Table A2 that the asymptotic standard errors are not badly estimated, even if a larger proportion of the fits have failed. We saw in Section 5 an example that was still more extreme.

**Table A1.** Theoretical values of the parameters, averages and standard deviations of the estimates across simulations and averages across simulations of the estimated standard errors $\phi_1'$, $\phi_1''$, $\phi_2'$ and $\phi_2''$ for the tdAR(2) model described above for $n = 400$ and 999 replications (out of 1000).

| Parameter | | Standard | Average of |
|---|---|---|---|
| True Value | Average | Deviation | Standard Error |
| $\phi_1' = 0.0$ | 0.007306 | 0.050587 | 0.043869 |
| $\phi_1'' = 0.002551$ | 0.002422 | 0.000322 | 0.000333 |
| $\phi_2' = -0.2$ | $-0.193960$ | 0.048853 | 0.043537 |
| $\phi_2'' = 0.003571$ | 0.003421 | 0.000332 | 0.000325 |

**Table A2.** Theoretical values of the parameters, averages and standard deviations of the estimates across simulations and averages across simulations of the estimated standard errors $\phi_1'$, $\phi_1''$, $\phi_2'$ and $\phi_2''$ for the tdAR(2) model described above for $n = 50$ and 934 replications (out of 1000).

| Parameter | | Standard | Average of |
|---|---|---|---|
| True Value | Average | Deviation | Standard Error |
| $\phi_1' = 0.0$ | 0.01419 | 0.14260 | 0.13620 |
| $\phi_1'' = 0.020408$ | 0.01640 | 0.00900 | 0.00957 |
| $\phi_2' = -0.2$ | $-0.19510$ | 0.13972 | 0.12436 |
| $\phi_2'' = 0.028571$ | 0.02355 | 0.00852 | 0.00749 |

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
