# Peer review of "Autoregressive Models with Time-Dependent Coefficients—A Comparison between Several Approaches"

_stats, doi:10.3390/stats5030046_

Round 1

Reviewer 1 Report

In the abstract:

We do not say heteroscedastic variance, we should say instead heteroscedatic errors.

It is not clear what you mean in "we consider our theory in that case". What do you mean? In the abstract it is necessary to clearly present what was done in the paper.

Also, to say that the theory of application can play a role is very vague and it is not useful in this abstract.

What is alpha_{t} in (2.7) and (2.8)? It was not mentioned before.

Please, fix the y-axis of Figure 3.

I think that it is hard to identify how autoregressive parameters are changing over time.

Author Response

See my answer to all the reviewers

Reviewer 2 Report

This paper intends to provide a comprehensive review on autoregressive models with time-dependent coefficients and I do believe it completely fulfills its goals. The authors provide a thorough description of the state of the art and focus on the particular case of time-dependent autoregressive (tdAR) models, from the broader class of autoregressive-moving average (ARMA) models with time-dependent coefficients and marginally heteroscedastic innovation variance. For the particular tdAR case, the authors present the theory already developed in a previous work (Reference [1]) and extend some results, discuss previous topics and provide simulations to better illustrate their findings. They also present two alternative (and sometimes complementary) theories for tdAR models, namely the Dahlhaus theory for locally stationary processes, and the Bibi and Franck theory, with particular focus on the cyclically time-dependent models. Examples are presented to illustrate the different approaches, together with a discussion about the importance the field of application may have in the choice of the most suitable approach and all the three alternative theories are compared and their asymptotics discussed.

The results are clearly presented, and the paper is very well written and is easy to follow. I found it very interesting to review this paper and was very happy to have the opportunity to review it. I only have a few comments/suggestions regarding text editing that may help in enhancing paper's readability, which are the following: 

1. Please review spacing between lines because it seems it is not the same throughout the paper. It may sometimes be due to mathematical characters in text, but there are places (like page 16, for instance) where that is not the case.

2. Why is the first footnote numbered as "3", in page 5. Is there a reason for that?

3. Please consider rewriting line 135: is there the need to repeat " uniformly in n"?

4. Is sentence in lines 175 and 176 complete? It seems there is at least a word missing after "a new proof of consistency and asymptotic...".

5. Line 287 seems to continue legend of Figure 2. This may not be the final layout of the paper but, in this format, more space is needed between lines 286 and 287.

6. Please consider splitting sentence from lines 288 to 291: it is too long and becomes hard to follow.

7. In line 330, I believe it should be plural in "standard error".

Author Response

See my answer to all the reviewers

Reviewer 3 Report

In this paper, authors proposed autoregressive-moving average (ARMA) models with time-dependent (td) coefficients and marginally heteroscedastic in-novation variance provide a natural alternative to stationary ARMA models. In addition, there are also differences in terms of asymptotics as shown by an example. Their opinion is that the field of application can play a role here. Finally, the paper is completed by simulation results that show that the asymptotic theory can be used even for short series (less than 50 observations).

I suggest that (i) authors need to explain the practical applications for td-ARMA; (ii) Can authors give some motives for the td-ARMA?; (iii) authors also need to give some practical examples (not simulation examples) to explain their td-ARMA; (iv) the paper also needs a good proof-reading by a scientific writer. Hence, I feel that the main contribution of the paper is weak and the paper cannot be accepted for publication in this applied journal.

Author Response

See my answer to all the reviewers

Reviewer 4 Report

The paper is a nice piece of theoretical work on autoregressive models with time-dependent coefficients. The theoretical aspects are duly exposed and the results derived. Two aspects might be interesting for many readers:

1. in the reviewing the different methods, it would be nice if the authors might add some details about possible practical problems that have been treated using those methods;

2. readers might be interested in knowing more precisely if there is any piece of software that can deal with this kind of processes.

Author Response

See my answer to all the reviewers
